# Quantifying Nocturnal Scratch in Atopic Dermatitis: A Machine Learning Approach Using Digital Wrist Actigraphy

**DOI:** 10.3390/s24113364

**Published:** 2024-05-24

**Authors:** Yunzhao Xing, Bolin Song, Michelle Crouthamel, Xiaotian Chen, Sandra Goss, Li Wang, Jie Shen

**Affiliations:** 1Statistical Innovation Group, AbbVie, North Chicago, IL 60064, USA; 2Digital Science, AbbVie, North Chicago, IL 60064, USA; 3Department of Biomedical Engineering, Emory University, Atlanta, GA 30322, USA

**Keywords:** nocturnal scratch, digital health technology, wearable, machine learning, atopic dermatitis

## Abstract

Nocturnal scratching substantially impairs the quality of life in individuals with skin conditions such as atopic dermatitis (AD). Current clinical measurements of scratch rely on patient-reported outcomes (PROs) on itch over the last 24 h. Such measurements lack objectivity and sensitivity. Digital health technologies (DHTs), such as wearable sensors, have been widely used to capture behaviors in clinical and real-world settings. In this work, we develop and validate a machine learning algorithm using wrist-wearing actigraphy that could objectively quantify nocturnal scratching events, therefore facilitating accurate assessment of disease progression, treatment effectiveness, and overall quality of life in AD patients. A total of seven subjects were enrolled in a study to generate data overnight in an inpatient setting. Several machine learning models were developed, and their performance was compared. Results demonstrated that the best-performing model achieved the F1 score of 0.45 on the test set, accompanied by a precision of 0.44 and a recall of 0.46. Upon satisfactory performance with an expanded subject pool, our automatic scratch detection algorithm holds the potential for objectively assessing sleep quality and disease state in AD patients. This advancement promises to inform and refine therapeutic strategies for individuals with AD.

## 1. Introduction

Atopic dermatitis (AD) is a chronic skin condition characterized by intense itching, often in the evening or nighttime, which can disrupt sleep and negatively impact the quality of life for patients [1,2,3]. Traditional assessments of itching and sleep in AD rely on patient-reported outcomes (PROs), which may be subject to recall bias and subjectivity and, therefore, not accurately reflect the severity of nocturnal scratching [4]. To better quantify nocturnal scratching in AD patients, there is a need for a more objective measurement.

Previous studies have explored different approaches for objectively measuring scratching behavior, including finger flexion [5], sound detection [6], and actigraphy assessment of sleep quality and quantity [7,8]. However, none of these methods have demonstrated sufficient accuracy and practicality for routine use in clinical trials. Therefore, there is still an unmet medical need for a reliable and practical method of quantifying scratching in AD.

Wearable sensor technologies, such as wrist-worn actigraphy devices, have shown promise in objectively measuring and assessing various health-related factors [9,10,11]. These devices are cost-effective, noninvasive, and user-friendly, making them ideal for continuous health monitoring and the early detection of symptoms. In recent studies, actigraphy devices have been used to detect nocturnal scratching, but there is still variability in performance and limited evaluation of specificity due to the imbalanced dataset [12]. In addition, accurately identifying hand and finger scratch activity from non-scratch movement is also a challenging problem.

To address the limitations mentioned above, several studies have been conducted using machine learning (ML) and deep learning (DL) methodologies. For instance, Ji et al. employed a combination of commonly used time and frequency domain features, topological features, and deep learning extracted features. They trained a LightGBM machine to predict nocturnal scratches [13]. Similarly, Moreau et al. utilized raw actigraphy data directly to train recurrent neural networks (RNN) and long short-term memory (LSTM) models for the same purpose [14]. Additionally, Mahadevan et al. proposed an innovative hierarchical paradigm that incorporates a binary ML classifier. Their approach assesses nighttime scratching and sleep using accelerometer data from a wrist-worn wearable device [15]. These studies aim to improve the accuracy and effectiveness of predicting and evaluating nocturnal scratches and sleep by leveraging advanced ML and DL techniques.

In this work, we developed two types of machine learning models for scratch prediction using wrist-wearing actigraphy data and made thorough comparisons. The first model utilizes a feature engineering approach, employing extracted physical, interpretable features to train a gradient-boosting classifier. Simultaneously, a deep learning model is constructed, employing a convolutional neural network (CNN) to directly predict scratch from raw actigraphy data. By comparing the performance of both models, the advantages of each approach are identified. Additionally, this study explores the impact of various datasets, such as comparing the model performance using data from both hands versus data from a single (scratch-dominant) hand and comparing accelerometer data with gyroscope data. This comprehensive analysis enhances the understanding of AD and contributes to the advancement of drug development processes by enabling more effective clinical trials and monitoring of treatment efficacy.

## 2. Data and Methods

### 2.1. Data

Data from actigraphy and videography were collected during a clinical study (NCT04262791). For this research, information from seven subjects in the study was utilized, consisting of two healthy volunteers and five Atopic Dermatitis patients from two of the three sites. Two Apple Watches (Series 5, Watch OS6) [16] were used to gather wrist actigraphy data from both hands during sleeping time for the enrolled subjects. In the meantime, the video data were collected using a high-definition infrared night vision camera placed above the participants’ beds, and the data were manually annotated to identify scratching events. Using the Pruritus Numerical Rating Scale 7-Day version (NRS-7) at the screening and Day 1 visits, AD patients were categorized into AD-high (NRS-7 ≥ 4) and AD-low (NRS-7 ≥ 2) groups. The participant pool consisted of three AD-high patients, two AD-low patients, and two healthy volunteers with an average age of 27.4 (Table 1). Data were obtained over the course of two nights during which subjects slept in the clinic. Each subject wore two Apple Watches, one on each hand, while overnight video recordings were taken to provide ground truth for nocturnal scratching.

The recorded video files underwent a motion detection algorithm to eliminate sections with no discernible motion from subsequent scratch annotation. For each video segment that contains motion, as identified by the motion detection algorithm, an annotator will label the movement as either a scratch, a non-scratch (such as rolling over or stretching), or noise (if the motion is not due to the subject, such as a dust particle floating in front of the camera). Before lying down in bed and after getting up in the morning, the subjects were instructed to clap twice with both hands to help align the timestamps of different data streams.

The Apple Watches collected actigraphy data from both accelerometer and gyroscope sensors at a sampling rate of 100 Hz. The continuous data were segmented into 3-s windows for feature extraction as established by Arnaud Moreau et al. [14] and matched with the corresponding video record labels based on their timestamps. Any time offsets between clocks were corrected using the clapping actions observed in the video and the corresponding signals in the accelerometer data. The dataset was filtered further using motion detection algorithms, which relied on a power-like measure (mean square of magnitude) represented by equation 1 for each sample segment. Only segments exhibiting motion were selected for model training. Equation (1) is defined as follows:(1)$$P=\frac{1}{n}\sum_{t=1}^{n}{a_{x}}^{2}+{a_{y}}^{2}+{a_{z}}^{2}$$

In Equation (1), *n* represents the number of samples within the segment, while *a_x_*, *a_y_*, and *a_z_* represent the acceleration signals along the respective axes. In this study, a threshold of 0.0005 was set to determine whether a segment exhibited motion. Segments with *p* values greater than 0.0005 were identified as having motion and were included in the model training process. A total of 13,632 segments exhibited motion, while 13,335 segments were associated with non-scratch activities, and 297 segments were linked to scratch activities. The segments with motion were further divided into train, validate, and test groups based on subject level. The training dataset consisted of five subjects; one subject was allocated for validation, and one subject was reserved for testing, as shown in Table 2.

### 2.2. Models

This study employed two modeling pipelines to classify whether motion was a result of scratch activity or non-scratch activity, as illustrated in Figure 1. On the right side is the traditional machine learning approach with feature engineering, which extracted explicit time series features and utilized them to train a gradient-boosting machine. On the left side is the deep learning pipeline, which trained 1D-CNN models using the raw accelerometer data.

#### 2.2.1. Feature Engineering Model

To extract features from the raw accelerometer and gravity data for the feature engineering model, the Python package *tsfresh* [17] was used. The *ComprehensiveFCParameters* were utilized as extractor settings, and median value imputation was applied. The fast Fourier transform (FFT) was applied to the data of all three axes (x, y, z) to extract the frequency domain features, which were added to the dataset. This process resulted in the extraction of 2353 features that quantified a range of time series characteristics, including autocorrelation, percentiles, energy, and peak frequency. Before training the model, all features were normalized by subtracting the mean and dividing by the standard deviation.

The prepared dataset was used to train a gradient-boosting machine from *scikit-learn* [18]. The hyperparameters of the model were tuned by evaluating its performance on the validation dataset. Additionally, during the hyperparameter tuning process, the optimal threshold for the probability cutoff was determined based on the performance of the validation dataset.

Specifically, the probability cutoff was defined as the intersection point between the precision and recall curves, as depicted in Figure 2. This approach aimed to strike a balance between precision and recall performance.

#### 2.2.2. Deep Learning (CNN) Model

The CNN model is trained on raw actigraphy data using a *ConvNormPool* architecture [19], comprising three layers of 1D convolutional layers, separated by 1D batch normalization layers, and a swish activation function [20]. This specific architecture is designed to extract meaningful signals from a time series of segment samples, with maximum pooling applied at the end. Three *ConvNormPool* modules are connected sequentially, followed by an average pooling layer and two fully connected layers, as depicted in Figure 1. 

The *ConvNormPool* architecture features a hidden size of 128 and a kernel size of 5. Adam optimizer is used, with an initial learning rate set at 0.001 and weight decay at 0.001. To manage the learning rate, a step learning rate scheduler is used with a step size of 10 and a gamma of 0.1. Focal loss is applied to loss criteria during training [21]. The model is trained for 200 epochs, utilizing a batch size of 16. Before initiating model training, reflection padding is applied to each sample segment to ensure alignment of the sample length in the time dimension. Throughout the training process, non-scratch (negative) samples are randomly selected to achieve a 1 to 1 sample size ratio with the scratch samples, thus ensuring sample balance. The model is trained using data from five training subjects, while the training progress is monitored using data from one validation subject. Finally, the model’s performance is evaluated using data from one test subject after the completion of model training.

## 3. Results

Table 3 presents the performance evaluation of both the feature engineering model and the CNN model. However, the accuracy metric is heavily influenced by the majority class (non-scratch) due to the significant data imbalance. This may not accurately reflect the performance at the scratch events. Therefore, in addition to evaluating precision and recall on the minority class (scratch), F1 scores are calculated by Equation (2) to provide a comprehensive evaluation that balances precision and recall. The results indicate that the feature engineering model exhibits superior performance in terms of the F1 score metric compared to the CNN model. Furthermore, our model achieves a comparable F1 score to the published work [13], but with less than half the number of subjects.
(2)$$F1=2\times \frac{precision\times recall}{precision+recall}$$

In addition to the accelerometer data, the gyroscope data are utilized independently to train the feature engineering model. The performance results are depicted in Table 4, revealing that the gyroscope model exhibits lower performance compared to the accelerometer model. Despite having the same recall, the model trained exclusively on gyroscope data demonstrates lower precision and F1 scores. Moreover, the F1 score of the gyroscope + accelerometer model experiences a slight drop when compared to the accelerometer model. This could potentially be due to the information captured by the gyroscope and accelerometer being similar or redundant, which may not offer significantly more information compared to using only the accelerometer data. It is worth noting that incorporating more features could introduce additional noise and complicate the model’s ability to select the most optimal features for the base learners rather than enhance performance. Additionally, the feature engineering model is tested using only the scratch-dominant hand’s accelerometer data, and its performance is compared with that of using data from both hands, as illustrated in Table 4. Notably, the results demonstrate that utilizing data from both hands yields better performance compared to using data from the scratch-dominant hand only.

The feature engineering model employs the *tsfresh* package to generate three types of features. These include features that quantify the unpredictability of fluctuations in a time series, features that quantify quantile-based characteristics of time series measures, and features that quantify the frequency components of time series measures. To gain further insights into the feature engineering model, the impurity-based feature importance is examined. Figure 3 illustrates that all three of the top features exhibit statistically significant differences between scratch and non-scratch activities.

## 4. Discussion

One of the primary challenges in modeling nocturnal scratch detection is the highly imbalanced nature of the data. In machine learning, an imbalanced training dataset can lead to a biased model that tends to predict the majority class. Consequently, this can result in low recall when predicting the minority class. In addressing the challenge of an unbalanced dataset in our study, we implemented a two-pronged strategy to enhance the robustness and accuracy of our machine learning models. Firstly, we redefined the classification problem from distinguishing between “scratching vs. non-scratching” actions to differentiating between “scratching movement vs. non-scratching movement” [13,15]. This approach significantly increased the ratio of positive samples, thereby enriching our dataset with more relevant positive samples for model training. Secondly, during the model training phase, we adopted specific techniques tailored to each model type to mitigate the imbalance issue further. For the convolutional neural network (CNN) model, balanced sampling was employed to ensure an equal representation of both classes in the mini-batch for each training iteration. In contrast, for the gradient-boosting machine (GBM) model, we utilized sample weighting as a method to give more importance to the underrepresented class. These strategies collectively aim to improve model performance by effectively dealing with the challenges posed by the unbalanced dataset.

In this study, we also examined the differences between using both hands and solely the dominant hand to capture the scratching behaviors. Previous studies have predominantly focused on both hands; however, it is important to evaluate whether the dominant hand alone would offer the most relevant insights for scratching behaviors in AD patients. To facilitate this comparison, the accelerometer data of the scratch-dominant hand were utilized to train the feature engineering model. The findings indicated that using the scratch-dominant hand alone can moderately capture scratching behavior, albeit with compromised precision.

In order to provide a valuable and comparative assessment between the feature engineering GBM model and the CNN model, both models were trained on the same dataset using standardized preprocessing techniques. Within the scope of the study’s current limited dataset and preprocessing methods, the feature engineering GBM model outperformed the CNN model. This outcome can be attributed to several factors that are crucial in the context of our dataset and the models’ architectural differences. Firstly, the smaller sample size inherent in our dataset may have favored the GBM model, which is typically more efficient at handling limited data without overfitting, compared to the CNN model, which often requires a large volume of data to generalize effectively. Furthermore, the feature engineering approach allowed for the explicit incorporation of domain knowledge, leading to a more informed and potentially more relevant feature set for the GBM model. Additionally, the CNN model’s complex hyperparameter search space poses a significant challenge, particularly with limited data. The process of fine-tuning a CNN involves navigating through a vast array of architectural choices and hyperparameters, which can be both time-consuming and computationally expensive, especially without a large dataset to validate the efficacy of these choices. In contrast, the GBM model, with its more straightforward parameterization and the ability to leverage engineered features, can often achieve comparable or superior performance with less effort on hyperparameter optimization. The comparison results indicate that customizing approaches to the specific characteristics of datasets could potentially enhance model performance. 

The main limitation of this study is that we did not reach our enrollment goal due to a direct consequence of the COVID-19 pandemic. The research site had to be closed due to pandemic-related circumstances, and logistical challenges as well as funding constraints make it unfeasible to reopen the trial for further participant recruitment. The limited number of subjects enrolled in this study not only affects the generalizability of our findings but also restricts the diversity of data captured. One potential consequence of the limited size of our dataset is that it may have affected the performance of the CNN model. Deep learning models, like CNN, typically demonstrate optimal performance when trained on larger datasets that provide a wider range of patterns and variations. Additionally, the use of wrist-worn sensors, while offering a non-intrusive and accessible means of data collection, inherently limits the scope of detectable activities. Specifically, these sensors may fail to capture events that involve only finger movements, such as subtle scratching actions, due to their location on the body. This limitation highlights a critical area for future research and development, emphasizing the need for advanced sensing technologies or methodologies capable of capturing a specific niche with greater precision.

Moreover, capturing certain rare cases may present challenges due to the position of scratching. Typically, the position of scratching has a minimal impact on detection results in most scenarios because wrist-worn sensors primarily capture the movements of the hand performing the scratching action. The gross movement signals produced by the hand exhibit characteristic frequencies typically within a few hertz or less, which wrist-worn accelerometers can readily detect. However, certain rare cases might pose challenges, such as when a person scratches the opposite hand while keeping the scratching hand relatively stable. In this scenario, the subtle vibratory impulses that arise from the scratching action might be difficult to capture due to signal attenuation as they travel through the fingers and wrist.

## 5. Conclusions

This study demonstrates that both the feature engineering model and CNN model exhibit decent performance in identifying nocturnal scratch activity from actigraphy data. However, it should be noted that the current data may not be sufficient due to the limited number of subjects and nights spent at clinical sites. Scratch events are relatively rare compared to non-scratch events, and thus, the performance of both models can be enhanced with more extensive data. The comparison between models trained using single-hand data and both-hand data indicates that the mixture of both-hand data improves performance. However, it is also possible that the improved performance is a result of having more data. Furthermore, upon comparing the data from accelerometers and gyroscopes, it becomes apparent that accelerometer data deliver slightly better performance. This finding differs from a previously reported study [13], indicating the need for further exploration in the future. In addition, an accelerometer generally consumes less energy compared to a gyroscope, thereby conserving battery life when it is the only sensor used. Taking into account the typically improved battery life of accelerometer-based devices, this finding implies that potential devices equipped solely with accelerometers could offer equivalent, if not better, prediction performance along with longer battery life, thereby reducing burdens for patients. Further analysis of the feature engineering model reveals that several features extracted from accelerometer data exhibit strong signals for identifying nocturnal scratches from non-scratch activities. This discovery instills greater confidence in the automatic counting of nocturnal scratches using wearable devices.

## Figures and Tables

**Figure 1 sensors-24-03364-f001:**
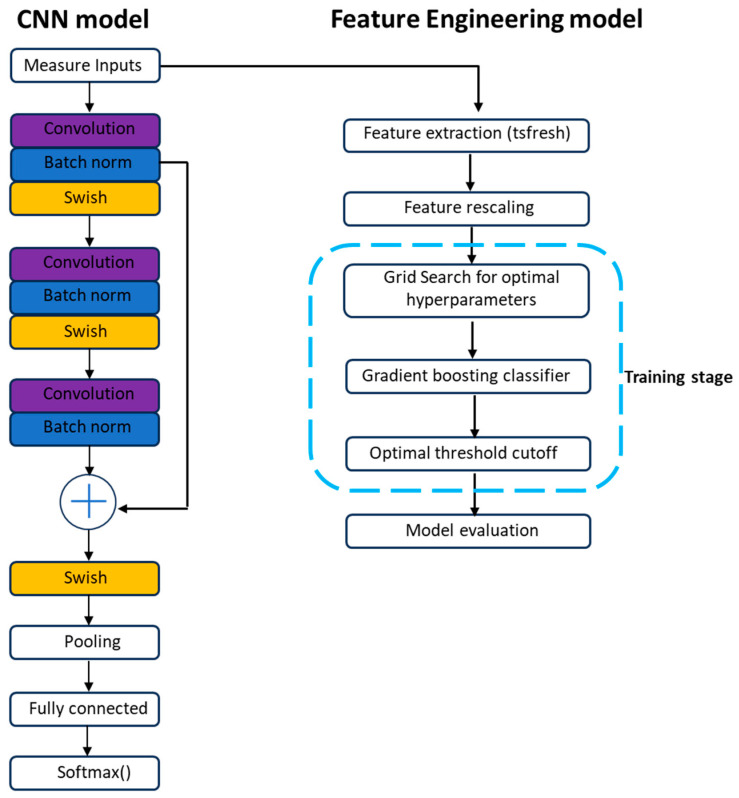
Model architecture illustration. Left side: CNN model; Right side: Feature engineering model.

**Figure 2 sensors-24-03364-f002:**
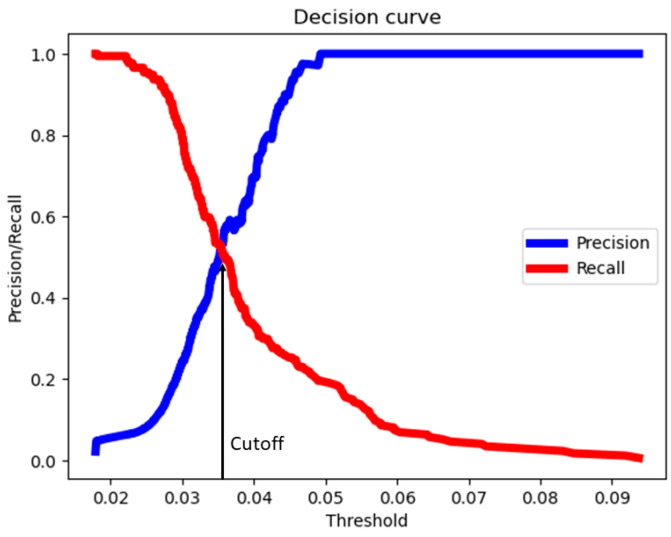
Decision curve to determine probability cutoff.

**Figure 3 sensors-24-03364-f003:**
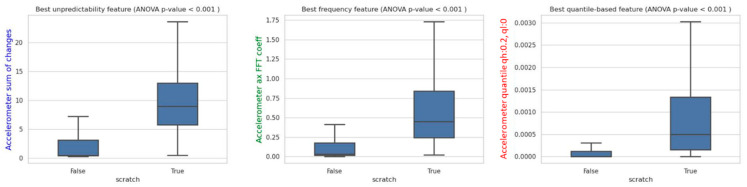
Wilcoxon rank-sum test on top three features in feature engineering model.

**Table 1 sensors-24-03364-t001:** Demographic information for all subjects.

Masked ID	Age	Sex	Race	AD Category
1	36	Male	Black/African American	Healthy
2	24	Male	Asian	AD-high
3	22	Male	Asian	AD-high
4	19	Female	Black/African American	AD-low
5	27	Female	White	Healthy
6	38	Female	White	AD-high
7	26	Female	White	AD-low

**Table 2 sensors-24-03364-t002:** Counts of scratch and non-scratch segments.

Masked ID	Group	Total	Scratch	Non-Scratch
1	Tuning	4893	29	4864
2	Training	3586	69	3517
3	Training	2931	97	2834
4	Training	942	4	938
5	Training	382	3	379
6	Testing	643	94	549
7	Training	255	1	254

**Table 3 sensors-24-03364-t003:** Model performance.

	Feature Engineering Model		CNN Model		Ju Ji et al. (2023) [13]
	Train	Test	Train	Test	Leave-One-Out
Precision	0.48	0.44	0.56	0.29	-
Recall	0.58	0.46	0.62	0.63	0.64
Accuracy	0.97	0.84	0.93	0.72	0.78
F1	0.52	0.45	0.59	0.39	0.44

**Table 4 sensors-24-03364-t004:** Model performance with various datasets.

	Accelerometer		Gyroscope	Gyroscope + Accelerometer
	Both Hands	Scratch-Dominant Hand	Both Hands	Both Hands
Precision	0.44	0.29	0.23	0.32
Recall	0.46	0.43	0.46	0.49
Accuracy	0.84	0.64	0.70	0.77
F1	0.45	0.34	0.30	0.39

## Data Availability

Upon request and subject to review, AbbVie will provide the aggregated and raw data that support the findings of this study.

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
