# Peer review of "Quantifying Nocturnal Scratch in Atopic Dermatitis: A Machine Learning Approach Using Digital Wrist Actigraphy"

_sensors, 2024, doi:10.3390/s24113364_

Round 1
Reviewer 1 Report
Comments and Suggestions for Authors
1.How does the performance of the feature engineering model compare specifically to the CNN model in identifying nocturnal scratch activity?
2.What are the specific performance metrics that show accelerometer data outperforming gyroscope data in this context?
3.How does the rarity of scratch events compared to non-scratch events affect the models' performance?
4.What specific limitations are mentioned regarding the sufficiency of the current data set?
5.What are the potential advantages of using accelerometer-based devices for the automatic detection of nocturnal scratching?
6.I can’t see any experimentally measured circuit structures or presentation of specific equipment in this article.
Author Response
We would like to express our sincere gratitude to the reviewer for the valuable and insightful comments. Your expertise and constructive feedback have greatly contributed to the improvement of our manuscript. We truly appreciate the time and effort you have dedicated to thoroughly reviewing our work. The suggestions and recommendations have been instrumental in refining our research. Thank you once again for the reviewer's invaluable contribution to the advancement of our study.
- How does the performance of the feature engineering model compare specifically to the CNN model in identifying nocturnal scratch activity?
Thank you for your valuable comment regarding the comparison of the performance between the feature engineering model and the CNN model in identifying nocturnal scratch activity.
In Table 3, we can observe that each model displays its own strengths. To provide an overall assessment, we utilized F1 score (Equation) to provide a balanced estimation between precision and recall. According to that, the feature engineering model outperforms the CNN model.
This outcome is not surprising, as deep learning models, such as CNN, typically excel when trained on sufficiently large datasets. In our case, the performance of the CNN model may have been impacted by the limited size of our dataset. Given that, it is understandable why the feature engineering model performs better.
To address this discussion more explicitly, we have made the necessary changes in Section 3 of the manuscript as follows. These revisions aim to provide a clearer explanation of the factors contributing to the varying performances of the two models.
“Table 3 presents the performance evaluation of both the feature engineering model and the CNN model. However, the accuracy metric is heavily influenced by the majority class (non-scratch) due to the significant data imbalance. This may not accurately reflect the performance on the scratch events. Therefore, in addition to evaluating precision and recall on the minority class (scratch), F1 scores are calculated using an equation to provide a comprehensive evaluation that balances precision and recall. The results indicate that the feature engineering model exhibits superior performance in terms of the F1 score metric compared to the CNN model...”
- What are the specific performance metrics that show accelerometer data outperforming gyroscope data in this context?
Table 4 demonstrates that the gyroscope model exhibits inferior performance compared to the accelerometer model. Despite having the same recall, the model trained exclusively on gyroscope data demonstrates lower precision and F1 scores.
To further address this point, we have added a detailed explanation in the manuscript as follows:
“Despite having the same recall, the model trained exclusively on gyroscope data demonstrates lower precision and F1 scores”
- How does the rarity of scratch events compared to non-scratch events affect the models' performance?
we acknowledge that one of the main challenges of nocturnal scratch detection modeling is the highly imbalanced nature of the data. In machine learning, an imbalanced training dataset can lead to a biased model that tends to predict the majority class. To conquer the problem due to the imbalanced dataset, we implemented a two-pronged strategy to enhance the robustness and accuracy of our machine learning models. Firstly, we redefined the classification problem from distinguishing between "scratching vs. non-scratching" actions to differentiating between "scratching movement vs. non-scratching movement." This approach significantly increased the ratio of positive samples, thereby enriching our dataset with more relevant positive samples for model training. Secondly, during the model training phase, we adopted specific techniques tailored to each model type to mitigate the imbalance issue further. In the GBM model, we give the positive samples more weight. For CNN model, we ensure the mini-batch for each training iteration is balanced.
We have updated the discussion section of our manuscript to provide a more detailed explanation of our approach to handling the imbalanced dataset. We believe that our strategy effectively addresses this challenge and improves the overall performance of the models. The following discussions have been added to the revised manuscript.
“One of the primary challenges in modeling nocturnal scratch detection is the highly imbalanced nature of the data. In machine learning, an imbalanced training dataset can lead to a biased model that tends to predict the majority class. Consequently, this can re-sult in low recall for predicting the minority class… For the convolutional neural network (CNN) model, balanced sampling was employed to ensure an equal representation of both classes in the mini-batch for each training iteration training process. In contrast, for the gradient boosting machine (GBM) model, we utilized sample weighting as a method to give more importance to the underrepresented class…”
- What specific limitations are mentioned regarding the sufficiency of the current data set?
The main limitation of this study is that we did not reach our enrollment goal, a direct consequence of the COVID-19 pandemic. The limited number of subjects enrolled in this study not only affects the generalizability of our findings but also restricts the diversity of data captured. One consequence is that the performance of the CNN model may have been impacted by the limited size of our dataset. The data limitation has been extensively discussed in the revised manuscript as following:
“The limited number of subjects enrolled in this study not only affects the generalizability of our findings but also restricts the diversity of data captured. One potential consequence of the limited size of our dataset is that it may have affected the performance of the CNN model. Deep learning models, like CNN, typically demonstrate optimal performance when trained on larger datasets that provide a wider range of patterns and variations”
- What are the potential advantages of using accelerometer-based devices for the automatic detection of nocturnal scratching?
An accelerometer normally consumes less energy than a gyroscope and therefore can save battery life if only the accelerometer is used. Taking into account the typically improved battery life of accelerometer-based devices, our finding implies that potential devices equipped solely with accelerometers could offer equivalent, if not better, prediction performance along with longer battery life, thereby reducing burdens for patients.
The potential benefits of using accelerometer-only devices were added to the Conclusions section as following.
“…it becomes apparent that accelerometer data delivers slightly better performance. An accelerometer generally consumes less energy compared to a gyroscope, thereby conserving battery life when it is the only sensor used. Taking into account the typically improved battery life of accelerometer-based devices, this finding implies that potential devices equipped solely with accelerometers could offer equivalent, if not better, prediction performance along with longer battery life, thereby reducing burdens for patients.”
- I can’t see any experimentally measured circuit structures or presentation of specific equipment in this article.
In this study, we used two commercially available Apple Watches (Series 5, Watch OS6) to collect wrist actigraphy data from both hands during our subjects' sleep periods. The Apple Watches used in our study are well-established and widely used devices in the consumer market, equipped with built-in motion sensors that can accurately capture and record wrist movements. In the revised manuscript, we added the reference to Apple Watch Series 5 technical specs (https://support.apple.com/en-lamr/118453)

Reviewer 2 Report
Comments and Suggestions for Authors
This paper meant to detect the nocturnal scratch in atopic dermatitis with the help of data from wrist actigraphy and machine learning methods. The work is useful in practical, but I do not think it is novelty enough for publishing in a scientific journal. The following is my main comments.
1.The motivation/novelty is vague. The detection of nocturnal scratch has been studied through many different methods, and the published works also achieved favorable results. I cannot get the reasons for the authors to conduct a new work. The mentioned methods and results contain no obviously sparks to distinguish the paper from others.
2. The given results should be compared with some other recently published ones to prove the superiority of your work, not only comparing with the two ways in the paper.
3. As the authors mentioned, the dataset is still strong enough for CNN to get a better classification results. More data should be collected from more AD patients to get a more comprehensive feature in the nocturnal scratches.
4. How to define and get the value of accuracy in Table 3?
5. The results in Table 4 shows the utilization of Gyroscope + Accelerometer leads to a drop for the performance. Why?
Author Response
Response to Reviewer 2
We would like to extend our heartfelt appreciation to the reviewer for the thoughtful and thorough evaluation of our manuscript. Your expertise and attention to detail have been instrumental in enhancing the quality and clarity of our research. We are grateful for the valuable comments and suggestions, which have significantly strengthened our work. Your insightful feedback has guided us in addressing important aspects and improving the overall coherence of our study. We extend our sincerest thanks for the time and effort invested in reviewing our manuscript, as the contributions have been invaluable to the advancement of our research.
- The motivation/novelty is vague. The detection of nocturnal scratch has been studied through many different methods, and the published works also achieved favorable results. I cannot get the reasons for the authors to conduct a new work. The mentioned methods and results contain no obviously sparks to distinguish the paper from others.
Thank you very much for your insightful comments and the time you have dedicated to reviewing our manuscript. We appreciate your feedback regarding the perceived novelty and motivation behind our study. We acknowledge that numerous studies have explored the detection of nocturnal scratching using various methods, achieving favorable results. We have conducted thorough literature review and summarized in the introduction section, such as, researchers have utilized deep learning models, such as LSTM, to predict nocturnal scratching (e.g., Arnaud Moreau et al., 2018). Additionally, others have employed machine learning models for the same purpose (e.g., Ju Ji et al., 2023). However, to the best of our knowledge, previous studies did not compare the performance of deep learning and machine learning models on the same dataset. In our study, we trained both a deep learning model (CNN) and a machine learning model using the same set of data to compare their performance. The result may provide valuable insights and guidance for future research in this area. Furthermore, most previous studies have used data from both hands (e.g., Ju Ji et al., 2023; Nikhil Mahadevan et al., 2021). It is worth investigating whether data from a single hand can yield comparable results to using data from both hands. Although Arnaud Moreau et al. (2018) explored the impact of active wrist versus both wrist movement data, the active wrist data still consisted of combined segments from both hands. In our study, we examined and compared the results obtained from data collected from a single (dominant) hand versus data from both hands. These findings could be instrumental in making decisions regarding future clinical trials, potentially providing economic benefits to sponsors and reducing burdens on patients by utilizing a single device.
- The given results should be compared with some other recently published ones to prove the superiority of your work, not only comparing with the two ways in the paper.
Thank you for your valuable comment. We have now compared our results with other recently published works, specifically the study by Ju Ji et al. (2023). Our model achieves a similar F1 score to their work, despite using less than half the number of subjects. We have included this comparison in table 3 and the results section to demonstrate the efficacy of our approach in relation to other relevant studies.
“Furthermore, our model achieves a comparable F1 score to the published work [13], but with less than half the number of subjects”
- As the authors mentioned, the dataset is still strong enough for CNN to get a better classification results. More data should be collected from more AD patients to get a more comprehensive feature in the nocturnal scratches.
We appreciate the reviewer's comments regarding the potential benefits of collecting more data to strengthen the classification results of our CNN model. However, as we noted in the manuscript, our data collection efforts were impacted by the COVID-19 pandemic, which prevented us from reaching our initial enrollment goal. Consequently, the number of subjects in our study was limited, which may have affected the generalizability of our findings and the diversity of our data. Nevertheless, we believe that the number of samples we were able to collect for most subjects was sufficient to draw preliminary conclusions, as we have presented in our manuscript.
- How to define and get the value of accuracy in Table 3?
Thanks a lot for the valuable comments. The estimation of accuracy in Table 3 relies on evaluating the model's performance using the test data set. This data comprises both scratch segments and non-scratch segments, and the evaluation metric used is the overall accuracy of the model's predictions without considering any weights. However, due to the significant imbalance in our data, the accuracy metric is heavily influenced by the majority class (non-scratch), potentially causing bias. To counter this, we have included additional metrics such as precision, recall, and F1 score, which offer an unbiased evaluation of the model's performance.
We have included the equation for F1 (equation 2) and an explanation in the Results section.
“Table 3 presents the performance evaluation of both the feature engineering model and the CNN model. However, the accuracy metric is heavily influenced by the majority class (non-scratch) due to the significant data imbalance. This may not accurately reflect the performance on the scratch events. Therefore, in addition to evaluating precision and re-call on the minority class (scratch), F1 scores are calculated by Equation 2 to provide a comprehensive evaluation that balances precision and recall. The results indicate that the feature engineering model exhibits superior performance in terms of the F1 score metric compared to the CNN model. Furthermore, our model achieves a comparable F1 score to the published work [13], but with less than half the number of subjects.”
- The results in Table 4 shows the utilization of Gyroscope + Accelerometer leads to a drop for the performance. Why?
We appreciate the reviewer's comments regarding the observed performance drop in the Gyroscope + Accelerometer model. Typically, having more data allows for more information to be captured, resulting in improved model performance. However, in the case of utilizing Gyroscope + Accelerometer, although it provides additional features, it does not increase the number of samples. Consequently, if the information captured by Gyroscope and Accelerometer is similar or redundant, it may not provide significantly more information compared to using just the Accelerometer data alone. Furthermore, rather than enhancing performance, incorporating more features could introduce more noise and complicate the model's ability to select the most optimal features for the base learner. We have included additional explanations in the revised manuscript as follows.
“Moreover, the F1 score of the Gyroscope + Accelerometer model experiences a slight drop when compared to the Accelerometer model. This could potentially be due to the information captured by the gyroscope and accelerometer being similar or redundant, which may not offer significantly more information compared to using only the accelerometer data. It is worth noting that incorporating more features could introduce additional noise and complicate the model's ability to select the most optimal features for the base learners, rather than enhancing performance.”

Round 2
Reviewer 1 Report
Comments and Suggestions for Authors
Accept in present form
Author Response
Thank you for reviewing the manuscript and shaping it to a much better position
Reviewer 2 Report
Comments and Suggestions for Authors
Thank you for taking my comments, but I’m still wondering about the achievements and purpose of this paper.
1.The authors claimed their contributions includes the comparison between DL and ML methods. However, the obtained results are greatly affected by the used model, target data and pretreatment, which cannot be a very generalizable guide for other results.
2. About the datasheet, I cannot get the great effects from the COVID-19 pandemic, which has been released hundreds of days before. There is still a long time between it and the submission.
3.About the novelty. As I mentioned before, the methods and their applications are not new or greatly improved and no important findings are provided. I cannot get an obvious spark to make the paper acceptable.
4. The performances of only Acc and Acc+Gyr is affected by the information redundancy and noise as the authors claimed. Please give an obvious evidence and get rid of this redundancy to improve the final detection results.
5. Will the scratching position influence the detection results? A discussion about it is suggested.
Before addressing all these issues, I still cannot give a positive recommendation for this paper.
Author Response
We extend our heartfelt thanks for your continued review of our manuscript and for sharing your additional insightful comments. Your thorough and thoughtful evaluation has been invaluable in helping us further refine our work. Below are our detailed responses to each of your concerns:
1.The authors claimed their contributions includes the comparison between DL and ML methods. However, the obtained results are greatly affected by the used model, target data and pretreatment, which cannot be a very generalizable guide for other results.
We acknowledge your point that comparisons between DL and ML methods may vary depending on the chosen models, target data, and pretreatment methods. Our goal was to offer a valuable and comparative assessment by training both models on the same dataset using standardized preprocessing techniques. We recognize that different datasets or preprocessing methods could yield varied results, which is why we presented our findings as comparative insights rather than definitive conclusions. We hope our results can guide researchers in choosing suitable approaches based on their specific dataset characteristics. In order to further clarify the message, we have added the following clarification in the revised manuscript:
“In order to provide a valuable and comparative assessment between the feature engineering GBM model and the CNN model, both models were trained on the same dataset using standardized preprocessing techniques. Within the scope of the study's current limited dataset and preprocessing methods, … The comparison results indicate that customizing approaches to the specific characteristics of datasets could potentially enhance model performance.”
- About the datasheet, I cannot get the great effects from the COVID-19 pandemic, which has been released hundreds of days before. There is still a long time between it and the submission.
Thank you for your concerns regarding the limitation of the sample size due to COVID-19. As we mentioned, this study was terminated prematurely due to the pandemic, and the research site had to be closed for this clinical trial. This made it impossible to reopen the trial for further recruitment of participants. Even though the pandemic has ended a while ago, reopening the study was not a feasible option due to logistical challenges and funding. However, we are confident that the collected data remains representative and valuable, even if the sample size is smaller than initially planned. We recognize the limitations this imposes on the generalizability of our findings and have added following clarification in the revised manuscript.
“The main limitation of this study is that we did not reach our enrollment goal due to a direct consequence of the COVID-19 pandemic. The research site had to be closed due to pandemic-related circumstances, and logistical challenges as well as funding constraints make it unfeasible to reopen the trial for further participant recruitment. The limited number of subjects enrolled in this study not only affects the generalizability of our findings but also restricts the diversity of data captured.”
3.About the novelty. As I mentioned before, the methods and their applications are not new or greatly improved and no important findings are provided. I cannot get an obvious spark to make the paper acceptable.
We understand your concerns about the perceived novelty of our methods and applications. However, we respectfully disagree with the assertion that "no important findings are provided." Besides offering a comparative analysis of different ML approaches, our study also explored the potential of monitoring only the dominant hand for scratch detection. This topic has been widely debated in the medical community, and no clear consensus has been established so far. Although our data is limited, our findings represent an important step forward in evaluating this critical issue, showing that monitoring the dominant hand alone could capture scratching activity with reasonable accuracy. This could significantly reduce the complexity of monitoring while still providing valuable insights into nocturnal scratching behavior. Additionally, our work contributes novel findings on the comparative effectiveness of the models, providing insights that could guide future research. We hope this clarification provides a better understanding of the importance of our work, and we greatly appreciate your continued review.
- The performances of only Acc and Acc+Gyr is affected by the information redundancy and noise as the authors claimed. Please give an obvious evidence and get rid of this redundancy to improve the final detection results.
We appreciate your insightful comment on redundancy and noise within the detection results. Our comparative analysis suggests that the combination of accelerometer and gyroscope data may introduce redundancy due to the significant overlap in motion signals captured by both sensors. This overlapping data, particularly in the frequency and motion domains, could interfere with the model's ability to identify relevant features effectively, thereby reducing overall performance. However, it is important to note that this redundancy hypothesis is yet to be fully confirmed, requiring further validation through additional studies. We proposed using only accelerometer data to mitigate this issue, as it delivers sufficient sensitivity to detect nocturnal scratching without introducing redundant or extraneous information. This approach simplifies the data collection process while reducing noise and preserving accuracy. Additionally, as we discussed in the manuscript, our findings do not align perfectly with previously reported results. This discrepancy highlights the need for more comprehensive research and provides novel insights into understanding optimal sensor usage for nocturnal scratch detection. We hope this detailed explanation clarifies our approach and emphasizes the unique perspectives gained from our study. We hope this detailed discussion clarifies our approach and underscores the rationale for focusing solely on accelerometer data in improving scratch detection results.
- Will the scratching position influence the detection results? A discussion about it is suggested.
We fully agree with your suggestion to further discuss whether the position of scratching could influence detection accuracy. After intensive debate and literature review, we believe that the position of scratching typically does not significantly influence detection results in most scenarios because the sensor primarily captures movement from the hand performing the scratching action. Therefore, variations in the location of the itch or the region being scratched usually have minimal relevance to the data collected. However, rare cases exist where scratching another hand while keeping the scratching hand stable can influence detection accuracy. For instance, if a subject scratches the opposite hand while maintaining minimal movement with the scratching hand, this behavior might result in a subtle signal that could be difficult to detect. Two types of signals are generally produced during scratching. The first type, which corresponds to gross hand movements, has characteristic frequencies in the range of a few hertz or less. The second type arises from subtle vibratory impulses generated by fingertip or fingernail motions against the skin or another surface. This signal can extend up to a few hundred hertz and attenuates significantly along the fingers and hand, with lower amplitudes at the wrist. This phenomenon is characterized by the decay of high-frequency signals as they propagate from the fingertip through the soft tissues of the hand and wrist. Given that our approach utilizes wrist-worn accelerometers, it primarily captures the first type of gross movement signal while having limited fidelity in detecting the higher-frequency second type. These gross movements, detectable by accelerometers operating at relatively low bandwidths (usually below 50 Hz), are suitable for capturing most scratching behavior. However, due to their distance from the fingertips and relatively lower bandwidth, wrist-worn sensors are less capable of capturing the second type of signal. Despite these limitations, our approach still provides reliable detection for the majority of scratching activities because most scratching movements involve visible hand and wrist motions. However, some rare cases could still be challenging to detect, such as scratching actions involving limited hand movement. Therefore, our findings are a valuable step forward in evaluating scratching detection and provide novel insights compared to previous studies. They reinforce the importance of optimizing the sensor location and configuration for comprehensive detection.
In the revised manuscript, we have added the following discussion:
“Moreover, capturing certain rare cases may present challenges due to the position of scratching. Typically, the position of scratching has a minimal impact on detection results in most scenarios because wrist-worn sensors primarily capture movements of the hand performing the scratching action. The gross movement signals produced by the hand exhibit characteristic frequencies typically within a few hertz or less, which wrist-worn accelerometers can readily detect. However, certain rare cases might pose challenges, such as when a person scratches the opposite hand while keeping the scratching hand relatively stable. In this scenario, the subtle vibratory impulses that arise from the scratching action might be difficult to capture due to signal attenuation as they travel through the fingers and wrist.”
